# Modelling the volcanic ash plume from Eyjafjallajökull eruption (May 2010) over Europe: evaluation of the benefit of source term improvements and of the assimilation of aerosol measurements

Matthieu Plu[1], Guillaume Bigeard[1], Bojan Sič[1], Emanuele Emili[2], Luca Bugliaro[3], Laaziz El Amraoui[1], Jonathan Guth[1], Beatrice Josse[1], Lucia Mona[4], and Dennis Piontek[3]

[1]CNRM, Université de Toulouse, Météo-France, CNRS, Toulouse, France
[2]CECI, Université de Toulouse, Cerfacs, CNRS, Toulouse, France
[3]Deutsches Zentrum für Luft- und Raumfahrt e.V. (DLR), Oberpfaffenhofen, Germany
[4]CNR-IMAA, Consiglio Nazionale delle Ricerche, Istituto di Metodologie per l'Analis Ambientale, Tito (PZ), Italy

**Correspondence:** Matthieu Plu (matthieu.plu@meteo.fr)

**Abstract.**

Numerical dispersion models are used operationally worldwide to mitigate the effect of volcanic ash on aviation. In order to improve the representation of the horizontal dispersion of ash plumes and of the 3D concentration of ash, a study was conducted using the MOCAGE model during the EUNADICS-AV project. Source term modelling and assimilation of different data were investigated. A sensitivity study to source term formulation showed that a resolved source term, using the FPLUME plume-rise model in MOCAGE, instead of a parametrised source term, induces a more realistic representation of the horizontal dispersion of the ash plume. The FPLUME simulation provides more concentrated and focused ash concentrations in the horizontal and the vertical dimensions than the other source term. The assimilation of MODIS Aerosol Optical Depth has an impact on the horizontal dispersion of the plume, but this effect is rather low and local, compared to source term improvement. More promising results are obtained with the continuous assimilation of ground-based lidar profiles, which improves the vertical distribution of ash and helps to reach realistic values of ash concentrations. Using this configuration, the effect of assimilation may last for several hours and it may propagate several hundred kilometres downstream of the lidar profiles.

## 1 Introduction

Volcanic ash is a potential threat to aircraft engines (Clarkson et al., 2016), and the atmospheric transport of ash clouds can cause severe perturbations and even disruptions to air traffic, and large economic losses (IATA, 2010). Continuous monitoring of ash clouds worldwide has been the duty of Volcanic Ash Advisory Centres (VAAC), that issue warnings and information in their respective domain of responsibility (ICAO, 2016). They provide at least qualitative information (i.e., presence of ash in different vertical layers, at different forecast lead times), and some VAACs also issue quantitative estimates of ash concentration. In Europe, the London and Toulouse VAACs issue messages when volcanoes erupt in their domain of duty, to warn of the presence of ash in different layers, defined as flight level (FL) bands: FL000-200, FL200-350, FL350-550. Since the Eyjafjaloküll eruption in 2010, it has been recognized that aircrafts may tolerate some ash ingestion and that

procedures should be revised (Bolić and Sivčev, 2011) in the sense that decisions of flying should be taken according to the tolerance of aircraft engines to ash concentrations. As a consequence, the London and Toulouse VAACs provide concentration charts (ICAO, 2016) for different thresholds: >0.2 mg.m$^{-3}$ and <2 mg.m$^{-3}$ (low contamination), >2 mg.m$^{-3}$ and <4 mg.m$^{-3}$
(medium contamination), and >4 mg m$^{-3}$ (high contamination). The concentration forecasts are given in the same FL bands as stated above, up up to 18 hours ahead.

In order to issue reliable forecasts, outputs from numerical dispersion models are widely used, combined with observations from satellites or ground-based stations. However, accurate forecasts of ash concentrations in near-real time remain a challenge, due deficiencies in numerical models, to lack of observation data, and to inherent uncertainties. Active research is on-going to
improve ash dispersion forecasts (Beckett et al., 2020), while remaining cost-effective to deliver data and warnings in a timely operational context. Some limitations in models arise from the insufficient resolution (horizontal, vertical, time step), from the representation of turbulence, of diffusion of the microphysical processes (aggregation, sedimentation) that account for the evolution of aerosols and of volcanic ash. The driving meteorological forecasts, for which error grows inevitably with time (Dacre et al., 2016), are also an important source of uncertainty for volcanic ash dispersion.
The volcanic source term, i.e. the mass of ash that is injected in the atmosphere as a function of height and time, is prone to large uncertainties and is another domain of active research. Different levels of complexity of source terms have been developed, which consist in deriving eruption parameters (mass eruption rate – MER, vertical profiles of injection of ash mass, grain size distribution, etc), from sparse and uncertain input measured data (plume height, ash columns, etc). Source terms also should also depend on the meteorological environment around the eruption. "Parametrised" source terms (such as Mastin
et al., 2009) provide values or analytical relationships between the eruption parameters based on past eruptions data; with the advantage of requiring very low computational resources. "Resolved" source terms are the result of an explicit simulation of the thermodynamic and buoyancy processes in the plume (such as the steady 1-D model PLUMERIA, Mastin, 2007), and even the microphysical aerosol processes, including aggregation (FPLUME, Folch et al., 2016). Source inversion of volcanic ash columns measured by satellites has also been developed in different institutes (Stohl et al., 2011; Steensen et al., 2017a;
Beckett et al., 2020). There purpose is generally to optimize a source term for which the model ash load columns match observed columns. Inversion requires an a priori that must be based on a parametrised or a resolved source term. Some studies have shown that the uncertainty of the result of inversion may be more linked to the uncertainty of the a priori than to the uncertainty of observations (Steensen et al., 2017b). Improving the physical representation of source terms is thus a critical topic.
One of the purposes of the European Natural Airborne Disaster Information and Coordination System for Aviation (EUNADICS-AV) project (Hirtl et al., 2019) was to develop and assess the integration of observations for air flight applications. Measurements from satellites and ground-based stations were considered for assimilation in dispersion models. Besides, some previous work have shown the benefits of assimilation of aerosol optical depth (AOD, Sič et al., 2016) and of lidar data (El Amraoui et al., 2020) for the 3D representation of aerosols. So, it is worth investigating whether the assimilation of AOD and of lidar
profiles may benefit to ash modelling, particularly when the ash cloud gets far from the volcano source.

The present article assesses the relative performance of different source terms and of the assimilation of satellite and ground-based data for the representation of 3-D concentrations of ash during a phase of the eruption of Eyjafjallajöküll in 2010. The experiments are done with the MOCAGE model and its assimilation scheme. MOCAGE is the model developed and used by Toulouse VAAC.

The plan of the article is as follows. Section 2 presents the MOCAGE model and the different observation datasets that are used in the case study. In Section 3, the different source terms are presented and their performance is compared. In Section 4, the assimilation of ground-based lidar data is presented and assessed compared to in-situ measurements of ash concentrations. The conclusion in Section 5 includes some perspectives of this work.

## 2 Case study

The study focuses on a particular period of the eruption of Eyjafjallajökull, from 13 to 20 May 2010. During this period, ash spread across the North Sea and the Atlantic Ocean, and impacted aviation operations over continental Europe (Fig. 1). The evolution of ash in the atmosphere was reported by several different observation systems (Section 2.2). Such measurements may be used for assimilation in MOCAGE (Section 2.1) and for model evaluation.

### 2.1 MOCAGE configuration

MOCAGE is a chemistry-transport model that is used for operations and for research at Meteo-France. The MOCAGE configuration that is used in the present study complies with the one described by Guth et al. (2016): it has full tropospheric and stratospheric chemistry, primary aerosols (desert dust, sea salts, volcanic ash, black carbon and organic carbon) and secondary aerosols (sulfate, nitrate, ammonium). The aerosols undergo various processes, as described by Guth et al. (2016): transport (advection and sub-grid transport), sedimentation, dry and wet deposition , and interaction with gas-phase chemistry. The aerosols are split into 6 size bins, that range from 2 nm to 50 $\mu$m with size bin limits of 2, 10, and 100 nm, and 1, 2.5, 10 and 50 $\mu$m. As an exception, the size representation of volcanic ash follow 6 $\phi$-scale classes (Krumbein, 1934), such as the ash MOCAGE bin 1 corresponds to the $\phi$-bins 10 and 9, bin 2 is $\phi$-bin 8, bin 3 is $\phi$-bin 7, bin 4 is $\phi$-bin 6, bin 5 is $\phi$-bin 5 and bin 6 is $\phi$-bin 4. This range of bins cover the size spectrum of fine ash (between diameter $2^{-4}$ mm = 62.5 $\mu m$) and diameter $2^{-10}$ mm $\simeq$ 1 $\mu m$), that can be transported over a long distance.

The MOCAGE simulations run on a global domain at 1° resolution, and on a large European domain at 0.2° resolution, called MACC02. The lateral boundary conditions for the smaller domain are provided by the global domain. The diagnostics are done on a subset of the European simulation domain (Fig. 1). Input meteorological forcings are provided with a 3-hours frequency: they come from ARPEGE 6-hourly analyses, interspersed with 3-hours forecasts. MOCAGE has 47 vertical hybrid sigma-pressure levels from the surface up to 5 hPa. The vertical resolution varies with altitude, with a resolution of 40 m in the planetary boundary layer, about 400 m in the free troposphere and about 700–800 m in the upper troposphere and in the lower stratosphere.

Assimilation of observations is done in the European MACC02 domain. The assimilation scheme in MOCAGE (Massart et al., 2009) relies on a variational incremental approach. From a model background $\mathbf{x^b}$ and a set of observations $\{y_i, i = 1..N\}$, it consists of minimizing the cost function

$$90 \quad \mathcal{J}(\delta x) = \mathcal{J}_b(\delta x) + \mathcal{J}_o(\delta x) = (\delta x)^T \mathbf{B}_t^{-1}(\delta x) + \sum_{i=1}^{N}(d_i - \mathbf{H}_i \delta x)^T \mathbf{R_i}^{-1}(d_i - \mathbf{H}_i \delta x) \, , \qquad (1)$$

where $\mathcal{J}_b$ and $\mathcal{J}_o$ are respectively the parts of the cost function related to the model background and to the observations, $\delta x = \mathbf{x} - \mathbf{x}^b$, $d_i = y_i - \mathbf{H}_i \mathbf{x}^b$, $\mathbf{H}_i$ is the observation operator for observation $y_i$, $\mathbf{B}$ is the background error covariance matrix, and $\mathbf{R_i}$ i is the observation error covariance matrix.

Assimilation applies in a hourly cycled continuous approach: the analysis at a given time is used as initial condition for the background one hour later. Two different variational methods may be used in MOCAGE: 3D-VAR or 3D-FGAT (first guess at appropriate time). Using an hourly 3D-VAR, the observations are assimilated at a hourly step and they are compared to the background at the same hour. Using 3D-FGAT (Massart S. et al., 2010), the same comparison steps apply, but in addition, an outer loop along an 3-hours assimilation window is assumed, that propagates back the increments to the beginning of the window.

For the assimilation of aerosols in MOCAGE (Sič et al., 2016; Descheemaecker et al., 2019; El Amraoui et al., 2020), the control vector $\mathbf{x}$ is the 3D total concentrations of aerosols. The choice of such control vector means that the column load and the vertical distribution of aerosols may be constrained by the assimilation, but the size and type distribution of aerosols will remain proportional to the distibutions in the background $\mathbf{x^b}$. Assimilation of multiple wavelengths is a possibility to constrain such distribution, but they have not been implemented yet.

The background error covariance matrix $\mathbf{B}$ influences the spread of the analysis to neighbouring grid boxes. It is specified with constant correlation lengths in the horizon- tal and the vertical. These correlation lengths may depend on the assimilation experiment. The observation error covariance matrix $\mathbf{R_i}$ is diagonal. The relative weights of variances given by $\mathbf{B}$ and $\mathbf{R}$ is important to specify the impact of observations on assimilation.

The observation operators $\mathbf{H}_i$ are needed for assimilation in order to translate the model state $\mathbf{x}$ into a simulated observation. These operators are described in the next section, for every kind of observations that may be assimilated.

## 2.2 Observations and observation operators

Several kinds of aerosol measurements are used in this study, either for assimilation in MOCAGE or for evaluation of the MOCAGE outputs. These observations are briefly presented here, together with the description of the AOD and lidar observation operators in MOCAGE assimilation.

### 2.2.1 VACOS ash concentrations from MSG/SEVIRI

The Volcanic Ash Cloud properties Obtained from SEVIRI algorithm (VACOS, Piontek et al., 2021b, a) derives volcanic ash coverage, ash optical thickness at 10.8 $\mu$m, mass column concentration, volcanic ash plume height and volcanic ash effective

particle radius from data of the passive SEVIRI imager aboard the geostationary Meteosat Second Generation (MSG) satellite. It consists of four artificial neural networks (ANNs) trained with a set of SEVIRI brightness temperatures calculated for a multitude of typical atmospheric settings including liquid and ice water as well as volcanic ash clouds using radiative transfer calculations. The ash optical properties were calculated for different refractive indices to cover the large variability of generic petrological compositions of volcanic ash (Piontek et al., 2021c). Besides the SEVIRI brightness temperatures in the thermal infrared, VACOS uses auxiliary data, including the satellite viewing zenith angle, the skin temperature from a NWP model and clear sky brightness temperatures derived from SEVIRI images.

VACOS has a fairly good volcanic ash detection probability (Piontek et al., 2021a) for ash layers with column loads between 0.2 and 1 $\mathrm{g\,m^{-2}}$ (between 1 and 10 $\mathrm{g\,m^{-2}}$) of approximately 93 % (99 %) and also allows for the quantification of the ash load of the plume with a mean absolute percentage error of ca. 40 % (26 %). The capacity of VACOS data to detect ash and to estimate ash load during this eruption phase has also been assessed by Plu et al. (2021). The overall conclusion is that VACOS can be reliably applied to detect volcanic ash concentrations larger than 0.2 $\mathrm{g.m^{-2}}$, and that the ash load estimates are in good agreement with estimates in the literature. The comparison with other satellite products shows similar peak values of ash load (3 $\mathrm{g\,m^{-2}}$ against 2 $\mathrm{g\,m^{-2}}$, Prata and Prata, 2012). Comparable ash loads have also been found at locations where lidar-based measurements have been done, e.g., around 1 $\mathrm{g\,m^{-2}}$ east of England on 17 May (Francis et al., 2012). So VACOS data may be used as a reasonable reference data set for assessing and comparing the performance of different model outputs. There is however underestimation on the high ash load values (above 10 $\mathrm{g\,m^{-2}}$) that can be retrieved. For such ash load, typical volcanic ash spectral signature in the thermal infrared might vanish if the ash plume becomes opaque (Watkin, Met. Appl., 2003). Besides, for simplicity, VACOS also neglects the impact of $SOi_2$ and ice-coated ash (Piontek et al., 2021b), both of which might be present close to the eruption source. These limitations apply in the denser parts of the plume, mostly close to the volcano.

### 2.2.2 MODIS AOD

The retrieved AOD values from the MODIS (Moderate-resolution Imaging Spectroradiometer) instruments onboard TERRA and AQUA (Levy and Hsu, 2015) can be assimilated in MOCAGE. Level-2 AOD at 550 nm (visible range) of the highest quality flag are considered: only pixels without any cloud contamination are kept. Since the MODIS AOD data have a higher horizontal resolution (10 km) than MOCAGE (0.2°), a super-observation approach is applied: at every hour and in every 0,2° grid cell, the mean value of all the observations that fall in this grid cell is used as the input for the assimilation. The observation operator for AOD in MOCAGE is described by Sič et al. (2016), except that the optical properties have been updated by Descheemaecker et al. (2019). Volcanic ash optical properties are taken from Pollack et al. (1973). The configuration of MOCAGE background error matrix $\mathbf{B}$ for the AOD assimilation experiments is as follows:

- the square-root of the background error variance is 30%,

- the horizontal correlation length is 2 gridpoints (ie, 0.4°),

- the vertical correlation length is two model levels.

An AOD observation error of 12% is assumed in the assimilation. These parameters follow the ones specified by Sič et al. (2016), except that the background error variance has increased, thus giving more weight to the observations than the background. Consistently with the assimilation scheme described in Section 2.1, the assimilation of AOD constrains the aerosol load but it does not constrain directly neither the vertical profiles nor the distributions of aerosol size and species, which proportions are kept as the ones in the background. The indirect effects (and improvements) of the AOD assimilation on aerosol vertical profiles are described by Sič et al. (2016).

### 2.2.3 EARLINET lidar profiles

The European Aerosol Research Lidar NETwork (EARLINET) was established in 2000, and it is now one of the components of ACTRIS (Aerosol Clouds Trace gases Research Infrastructure). In 2010, EARLINET investigated the spatio-temporal distribution of the Eyjafjallajökull emitted ash plume over European continent thanks to the almost continuous observations performed at its 27 lidar stations distributed over Europe (Pappalardo et al., 2013). A database devoted to reporting the geometrical and optical properties together with identification of the aerosol type for each of the aerosol layers observed during the whole related period is available at www.earlinet.org. Between 13 and 20 May, a significant ash layer was detected over Cabauw and Hamburg lidar stations. In the present study, the profiles of the aerosol backscatter coefficients at 532 nm (visible range) from these two lidars are assimilated in MOCAGE. Some ash load was also detected by a lidar located at Ispra, which will be used for evaluating the MOCAGE simulations.

The aerosol lidar observation operator in MOCAGE is similar to the one described by Janiskova and Stiller (2010). It offers the possibility to assimilate different retrieved variables: backscatter coefficients, extinction coefficient or attenuated backscatter profiles. The aerosol optical properties in the MOCAGE lidar observation operator are the same ones as for the MOCAGE AOD observation operator. The configuration of MOCAGE background error matrix $\mathbf{B}$ for the lidar assimilation experiments is as follows:

- the square-root of the background error variance is 50 $\mu$g.m$^{-3}$,

- the horizontal correlation length is 1° (e.g. roughly half of the distance between Cabauw and Hamburg),

- the vertical correlation length is two model levels.

An lidar backscatter observation error of 10% is assumed in the assimilation. The parameters have been inspired by the first design of lidar assimilation in MOCAGE by El Amraoui et al. (2020).

### 2.2.4 In-situ aircraft aerosol concentrations

Schumann et al. (2011) reported many research flights over continental Europe and the North Sea, during which in-situ measurements of ash concentration were taken. These measurements are important as they provide observations which can be directly compared to simulated ash concentrations. Although they are sparse in space and time (see Fig. 1), three flights (Flights

10, 11 and 12 from Schumann et al., 2011) will be used in the present study for the evaluation of 3D ash concentrations from the MOCAGE simulations.

## 3 Representation of the emissions and the plume

### 3.1 Sensitivity of dispersed ash to the source term

Many past studies have shown that ash dispersion is highly sensitive to the source term (Kristiansen et al., 2012; Steensen et al., 2017b; Beckett et al., 2020), and that it is necessary to describe as accurately as possible the mass eruption rate of the volcanic emission, the vertical distribution of ash aerosols in the column, and the particle size distribution. The complex processes and limited difficult-to-make observations of the volcanic plume source have driven, in the first place, the development of empirical parametrisations, as the attempt to define the emission term in models. Such parametrisations relate the height of 190 the eruption plume (as the parameter that can be readily be observed) and the mass of the eruption aerosols injected into the atmosphere. The usual and operational configuration of MOCAGE uses the Mastin et al. (2009) relation. Some of downsides of such parametrisations is that they do not address the question of the aerosol vertical distribution in the eruption column and they include only simplified description (if any) of the atmospheric conditions which influence the plume. Moreover, the height-mass relationship reflects a median behaviour based on past cases, and it is prone to important uncertainties from one 195 case to the other.

In order to overcome such limitations of empirical parametrisations, other approaches simulate physical processes within the plume and their interaction with the atmosphere. These so-called plume rise models are becoming increasingly sophisticated and can provide estimations of eruption and plume source parameters, such as the ejected mass and the particle vertical size distribution. The 1-D cross-section averaged plume rise model FPLUME (Folch et al., 2016) has been introduced in MOCAGE, 200 in order to assess the benefit of such plume rise model. FPLUME takes into account the effects of meteorological conditions on the thermodynamic of the plume, and of important physical processes like wet aggregation, air and particle entrainment and particle sedimentation. The FPLUME model is based on the turbulent buoyant plume theory. It resolves the height of an eruption plume from the eruption mass rate and the initial size distribution at the vent by solving the governing equations. It also outputs as a result the plume mass vertical distribution and the height-dependent particle size distribution for all vertical 205 levels within the plume. At the entry of FPLUME, a constant distribution of mass is supposed at the vent, which is the same as the one for the parametrised source term. FPLUME implements the Costa et al. (2010) aggregation model. FPLUME in MOCAGE takes into account the wind influence (from the meteorological fields given in MOCAGE) on the plume shape and height. At the output of FPLUME, ash is distributed into the corresponding MOCAGE ash size bins.

Two MOCAGE simulations are performed and compared: one with a parametrised empirical source term, the other with a 210 FPLUME resolved source term, for the Eyjafjallajökull eruption. In order to start the evaluation on 13 May from consistent initial ash concentrations, the emission starts from 9 May, 4 days before the period of evaluation. Plume height (Fig. 1) is taken from Arason et al. (2011), on which a simple pre-processing is applied: averaging is done at an hourly time step and at

**Table 1.** Input parameters and assumptions for the two MOCAGE simulations: the empirical parametrisation source term, and the FPLUME-resolved source term

| | Empirical parametrisation | FPLUME resolving |
|---|---|---|
| Plume top height | Input parameter (as described in the text and in Fig. 1b) | Input parameter (as described in the text and in Fig. 1b) |
| Physical assumptions of the volcanic eruption | None (useless) | Basaltic eruption type - Exit velocity (150 $m.s^{-1}$) |
| Total mass injected in MOCAGE | 30% of the total mass emitted, as recommended by Mastin et al. (2009) for medium-size silicic eruptions | Resolved by FPLUME (iterative mass solving, as Folch et al. (2016)) |
| Vertical mass profile | Uniform (from volcano vent up to the plume top) | Resolved by FPLUME |
| Aerosol size distribution | Uniform in the vertical, 6 $\phi$-bins: 10&9,8,7,6,5,4, with respective mass fraction (in %) 0.01,0.09,1.1,8.8,25,65 | Resolved by FPLUME in the vertical |

a 500m-accuracy height. This plume height information is used in both simulations to derive other source term parameters, as summarised in Table 1. In FPLUME, the MER is found by iterative solving (Folch et al., 2016) at every hour.

In FPLUME, the amount of particles that fall rapidly out of the plume (due too large size and mass) is very variable and it depends on the eruption type, initial size distribution, eruption phase and external meteorological conditions. In the present case using FPLUME, the percentage of the mass eruption rate of the ash particles that is dispersed (i.e. which size falls into the fine ash classes and will be introduced in MOCAGE) varies from 0.4% to 0.9%, depending on time. The effect of wet aggregation is rather low (less than 1%). In the case of the parametrised source-term, such variable effects cannot be produced

with realistic conditions: an empirical ratio of mass eruption rate of 30% is applied to account for the proportion of ash that is sufficiently fine to be dispersed, and the aerosol size distribution is uniform in time and vertical (Table 1). A 30% ratio of fine ash is recommended by Mastin et al. (2009) for a medium-size silicic eruptions, which corresponds to the case study.

  Time-altitude plots of the ash source term (Fig.2) point out how the different source terms can affect the MER and the vertical distribution of aerosol mass injection. Only fine ash that is then transported by MOCAGE is represented on the source

term plots. The MER are generally in a similar order of magnitude for both simulations, however, in the phases when the plume height is around 5000 m, the MER is generally higher for the FPLUME-resolved source term. When the plume reaches higher levels (around 8000m), on the contrary, the ash concentrations is generally higher for the parametrised source term. For the FPLUME-resolved source term, the highest concentrations of ash are in a layer of a few hectometers just above the neutral buoyancy level. Some ash mass is also emitted a few hundred of metres above the vent. For the parametrised source term, the

ash is homogeneously distributed between the vent and the maximum plume height, due to the prior assumption of uniform

vertical distribution of mass. Some tests have also been done using an umbrella-shape vertical distribution of ash mass, but the resulting atmospheric dispersion of ash was not better than a uniform vertical distribution.

In order to illustrate the horizontal dispersion of the plume from the two source terms, the simulations are compared to VACOS ash column load estimates, at three times (Fig. 3). In general, the model reaches higher values of ash load, which may be consistent with the underestimation of ash load above 10 g.m$^2$ that is discussed in Section 2.2.1. Besides, the model simulations represent continuous plumes, while the ash load retrieved from VACOS looks more discontinuous in space, with some isolated contaminated (resp. non-contaminated) gridpoints.

On 14 May at 06 UTC, a thin plume of ash has crossed the Atlantic and it reaches the Irish Sea and the Northern part of the British Isles. In both MOCAGE simulations, the plume has a realistic shape which goes in the right direction, compared to the ash plume seen in VACOS. On 16 May 2010 at 09 UTC, the plume has a similar direction but it is more horizontally extended than on 14 May. At both times, both MOCAGE simulations follow the VACOS plume shape, but the plumes in MOCAGE are thicker than the one detected by VACOS. The parametrised source term generates also areas with ash (off the coast of Ireland for instance, on 16 May at 09 UTC) which are not obvious in VACOS. The ash pattern that is west of Ireland in the parametrised simulation is mostly confined between the surface and 5 km altitude, which is below the denser plume (around 8km altitude). The most probable explanation is that the injection of mass at every vertical level in the parametrised simulation and not in FPLUME, combined with some vertical wind shear. On 17 May 2010 at 20 UTC, the shape of both plumes look also similar, with differences however near the volcano source and in the North Sea. A localized ash pocket aloft over Belgium and the Netherlands seen in VACOS does not show up in the simulations. Overall, the FPLUME-resolved source term generates a plume that it less spread out, which is consistent with a more vertically confined emission (Fig. 2). Indeed, in the presence of wind shear, different vertical distributions of ash can have large impact on the horizontal dispersion of ash load.

### 3.2 Impact of the assimilation of MODIS AOD

In order to evaluate the benefit of the assimilation of MODIS AOD for ash representation in MOCAGE, two additional simulations have been done, using respectively the two source terms. MODIS AOD data from AQUA and TERRA have been assimilated, using the configuration described in Section 2.2.2. Assimilation is done using the MOCAGE 3D-FGAT scheme at hourly step with a 3-hours window, continuously from the 10 May. Cumulative daily maps of the assimilated hourly values at 0.2° are shown in Fig. 4. In the areas where ash is present (between Iceland and the British Isles), many assimilated AOD gridpoints. On 14 and 16, some high AOD values belong to the plume and are presumably affected by volcanic ash.

Fig. 5 illustrates the effect of assimilation of MODIS AOD, by comparison to the simulations without assimilation (Fig. 3), at three times. On the simulations using the parametrised source term, the assimilation of MODIS AOD tends to limit the horizontal extent of the plume. On 16 May at 09 UTC, the ash plume off the Irish coast using the parametrised source term is mostly erased. On 17 at 20 UTC: ash load over Iceland diminishes after assimilation. On the FPLUME source term simulation, the effect of MODIS assimilation on ash load is less obvious, which may suggest that the AOD from this simulation agrees well with MODIS measurements. To summarize, the effect of the assimilation of MODIS on the horizontal extent of the plume is higher on the simulation with the parametrised source-term than on the FPLUME one. The effect of assimilation is mainly

to reduce ash load locally. In this first attempt to evaluate the impact of MODIS AOD on a volcanic ash plume, the impact is rather small. In a different context of a desert dust plume (Sič et al., 2016), assimilating AOD has a larger impact on the representation of the plume. The impact of the MODIS observations is a function of the number of observations covering the plume, and the impact may depend on the trajectory and the shape of the plume. In the following section, some metrics are shown to compare quantitatively the different simulations.

### 3.3 Mutual benefit of source terms and of assimilation

The evaluation of the different simulations is done against VACOS measurements, using a similar approach as Plu et al. (2021). VACOS and MOCAGE data are regridded at 0.2° resolution on the domain shown in Fig. 1. A gridpoint is considered to be contaminated by ash if ash load is above 0.2 g.m$^2$ (VACOS detection limit). Fig. 6 shows some diagnostics about the detection of ash by MOCAGE simulations compared to the VACOS measurements: hits (the number of contaminated gridpoints in both

MOCAGE and VACOS), false alarms (number of gridpoints that are contaminated in MOCAGE and not in VACOS), for all MOCAGE simulations. Detection is done on the same 0.2°-resolution grid, but the gridpoints where VACOS ash detection (due to meteorological water clouds for instance) was not possible are excluded from the analysis, even for the model outputs.

The time evolution of the number of contaminated gridpoints follows similar trends as the eruption evolves; for instance a maximum number of contaminated gridpoints is obvious some hours after the maximum phase of eruption (18 May at

280 00 UTC). However, the number of contaminated gridpoints for the model simulations is significantly higher than for the VACOS estimates. This is consistent with the examination of Fig. 3: the model contaminated areas are continuous, while the VACOS retrievals reveal the most contaminated areas.

The detection capacity (hit rates) of contaminated gridpoints is rather good for all models (second row of Fig. 6b), although there are different phases in the period considered. During the first phase of the eruption (from 13 to 16 May), a small number

of gridpoints are not detected as contaminated by the model simulations. Afterwards all contaminated gridpoints are correctly detected by simulations. Consistently with the evidence that the contaminated gridpoints in VACOS are lower than in the models, there is a high number of false alarms in all model simulations. However, it is noticeable that the number of false alarms is significantly lower for the FPLUME simulation than for the other source term. This is consistent with the fact that FPLUME generates a more condensed plume along the horizontal dimension (Fig. 3), remaining in better agreement with

the observed plume. Overall the assimilation of MODIS tends to diminish the false alarms without changing noticeably the detected area of ash. The impact of MODIS assimilation is lower for the simulation with FPLUME source term than for the parametrised source term.

The Fraction Skill Score (FSS) is a metric to assess the performance of volcanic ash dispersion simulations, by determining the scale over which a simulation has some skill (Harvey and Dacre, 2016) to locate ash plumes, according to a distance of

295 tolerance $r$. The implementation and use of FSS in this study is similar to Plu et al. (2021). It is calculated as:

$$FSS(r) = 1 - \frac{\sum_{j=1}^{N} \left[ O_j(r) - M_j(r) \right]^2}{\sum_{j=1}^{N} \left[ O_j^2(r) + M_j^2(r) \right]} \qquad (2)$$

with $N$ being the total number of gridpoints in the verification area, and $M_j(r)$ and $O_j(r)$ being the fractions of contaminated grid points within the circle of radius $r$ (in km-distance) around point $j$, for the model (MOCAGE simulation) and the observations (VACOS reference), respectively. Before the computation of $FSS(r)$, a normalization step was applied, where the $G$ most contaminated grid points were determined for VACOS and model data. For VACOS, all grid points (within the verification area) with ash load higher than $0.2\,\mathrm{g\,m^{-2}}$ are assumed to be contaminated; $G$ is defined as number of these grid points. For each model output, the $G$ grid points with the highest ash column load in the domain are kept for further analysis and used to calculate the FSS. This implies that a different set of $G$ grid points is derived compared to those determined from the VACOS data. After the normalization step, the FSS is a measure of the performance of the models to locate the most intense ash features, and it filters out the amplitude errors. A model has skill at a given scale when the FSS is above 0.5. The FSS can also be used to compare simulations: the higher FFS, the better. On Fig. 7, the FSS is shown for distance radius of 50km, 200km and 500km.

The FSS evolves in time following similar trends for all model simulations. For a distance of 50km, the FSS is not always above 0.5. When the radius increases, the score performs better: for a radius of 500km, the FSS is above 0.5 for all simulations most of the time. On the 19 May, the number of contaminated gridpoints in VACOS vanishes and the FSS descreases. It is noticeable that the FPLUME simulation has always better scores than the other source term. Besides, the assimilation of MODIS does not change the score at all times, but when it does, it is an improvement. The FSS metric confirms that the location of the plume using FPLUME is better than the other source term and that the assimilation of MODIS improves the location of the plume, but with an impact that is lower and less permanent.

## 4  Representation of the concentrations above Europe

In the previous section, the horizontal extent of the plume has been assessed for different numerical simulations. It has been shown that the FPLUME source term provides a better horizontal extension of the plume. However, the concentrations along the vertical dimension are an information that is also needed by air authorities. Plu et al. (2021) showed that the vertical distribution of ash is generally biased in source terms and dispersion models, at least on this case study. The purpose of this section is to assess simulations with regards to ground-based lidar measurements and to aircraft in-situ observations, between 17 May and 19 May, when the plume approaches and then spreads over continental Europe. In this section, the assimilation of lidar backscatter coefficients in the MOCAGE FPLUME configuration is assessed. MODIS AOD measurements are not assimilated in these experiments.

### 4.1  Assimilation of ground-based lidar profiles

Backscatter coefficients at 532 nm from the Cabauw and Hamburg lidars have been used in this study. The signature of ash can be seen on the backscatter profiles (Fig. 8a, d, g): high backscatter values may be seen around 4km on 17 May at 15 UTC at Cabauw, around 3km and 4.5km on 17 at 15 UTC at Hamburg, and around 4.5km on 18 at 9 UTC at Cabauw. The aerosol mask analysis developed for the aerosol lidar observations (Mona et al., 2012) identified as volcanic such lofted layers, but also as

mixed volcanic ash-local aerosol content in the lowest aerosol layers below the top of atmospheric boundary layer (Pappalardo et al., 2013). At the same instants, the MOCAGE simulation (without assimilation) shows different profiles of backscatter coefficient (Fig. 8b, e, h) and of ash concentrations (Fig. 8c, f, i): volcanic ashes reach rather high values (from 20 to 150 $\mu$g.m$^{-3}$), but the highest concentrations may be found in the lower levels, around 1 to 2 km altitude. Even though a mixing of ash and continental aerosols have been observed in the boundary layer (Pappalardo et al., 2013), the high concentrations of ash in MOCAGE in the lowest levels may also be due partly to some shortcomings in the representation of vertical mixing processes in the model (Plu et al., 2021), such as insufficient vertical resolution, grid-scale vertical velocity, diffusion, aerosol sedimentation.

The assimilation of lidar backscatter profiles in MOCAGE is done using 3D-VAR, using a continuous hourly cycle, from 17 May at 00 UTC until 19 May at 00UTC. Some pre-processing of the raw lidar profiles is needed, due to the fact that the vertical resolution of EARLINET backscatter profiles is much finer (100m) than the MOCAGE vertical resolution. In order to avoid inconsistencies in the assimilation process, the lidar profiles are regridded at a resolution similar to MOCAGE. Two different datasets are prepared for assimilation:

– "EARLINET mean": the assimilated value is the mean value of lidar backscatter coefficients between two MOCAGE levels,

– "EARLINET max": the assimilated value is the maximum value of lidar backscatter coefficients between two MOCAGE levels.

Such profiles ("mean" and "max") have been processed in order to assess the sensitivity of the assimilation to the pre-processing of lidar data. The high values of backscatter coefficients that can be seen in the lowest levels are kept for assimilation. Since the MOCAGE control vector includes all the types of aerosols that the model is able to represent (Guth et al., 2016), it is expected that the assimilation will split the contribution of continental aerosols and of ash according to the proportion in the model background.

The profiles corresponding to the assimilated data (Fig. 8) shows how the assimilation process behaves. The peaks of lidar backscatter coefficients at altitudes between 2 and 5 km correspond to the location of the ash cloud. Without assimilation, MOCAGE does not show a local maximum of ash at these locations, but rather a quasi-uniform distribution of ash between the surface and 6 km (at Cabauw), or a peak just below 2km (at Hamburg), consistently with the results of Plu et al. (2021). The simulations using assimilation of lidar profiles have higher concentrations of ash at the right altitude range. However, the peaks of backscatter coefficients and of ash concentration after assimilation are much smoother in the vertical compared to the assimilated lidar profiles (Fig. 8a, d, g). It is also obvious that the backscatter coefficients after assimilation (around 0.5 m$^{-1}$.sr$^{-1}$) are still much lower than the observation values that are assimilated (around 2 m$^{-1}$.sr$^{-1}$), which may be due to weight of the model background, to the model resolution and to the vertical error correlation. Assimilating mean or max lidar data generate similar shape of MOCAGE ash profiles, but they are highly different in amplitude.

The assimilated profiles on 18 May at 9 UTC over Cabauw look more consistent with the lidar profile than on 17 May. A possible explanation can be that the ash cloud has been assimilated continuously and longer in time on 18 May, at a time

when the corrections have been accumulated and propagated in time and in space. The assimilation of lidar backscatter profiles has also a large effect on ash concentrations in the boundary layer. On 17 May, the assimilation increases drastically the ash concentrations in the boundary layer. Since the increments of ash are linked to proportion of ash in the background $\mathbf{x}^{\mathrm{b}}$, if the proportion of ash in the background is too large, then the correction increases too much ash with regards to the other aerosols. It is probable that continental aerosols have a negative bias in MOCAGE (Descheemaecker et al., 2019), which has a double detrimental effect: it explains a negative bias of backscatter coefficients prior to assimilation, and it increases the proportion of ash in the boundary layer after assimilation.

## 4.2   Evaluation against in-situ aircraft measurements

During the period of the study, airborne measurements have been reported in the literature. Schumann et al. (2011) reported in-situ estimates of 3D ash concentrations above the North Sea, Germany, and the Netherlands on 13, 16, 17, and 18 May. Such aircraft measurements provide estimates of the ash concentrations at different levels, although with high uncertainty margins. In order to evaluate the benefit of assimilation of lidar profiles, comparisons of ash concentrations at the MOCAGE levels that correspond to the altitude of the aircraft measurements are provided.

The first flight considered is Flight 10 over "North Sea", on 17 May around 16 UTC (Fig. 9). The aircraft flew in a layer of ash between 3.2 and 6.3 km, where concentrations of ash between 105 and 283 $\mu$g.m$^{-3}$ were measured. In the MOCAGE levels at this instant, the assimilation of lidar data increases the concentration of ash, as shown on Fig. 9. The flight is quite close to the Cabauw lidar, and as shown in Fig. 8, the result of assimilation still leads to underestimation of ash concentrations. The core of highest concentrations of ash in the model are still located north to the flight measurements. The local concentration values (Fig. 11a) confirms that the assimilation has little impact at the flight location.

The ash concentrations in MOCAGE corresponding to two flights over continental Europe on 18 May around 10 UTC are examined on Fig. 10. The Flight 12 around Stuttgart measured ash concentrations between 16 and 38 $\mu$g.m$^{-3}$ at altitude 5.2km. In the MOCAGE simulation without assimilation, the plume has a thin shape and the concentrations around the flight (upper-panel of Fig. 10) are below 25 $\mu$g.m$^{-3}$. After assimilation, the MOCAGE simulations shows a clear increase of ash concentrations and of the extent of the plume, that covers a larger part of Germany. The ash concentration values around the flight are closer to the measurements after assimilation of lidar profiles (Fig. 11c). The assimilation of maximum lidar profiles fit well the measurements.

The flight 11 around Hamburg measured ash concentrations between 38 and 93 $\mu$g.m$^{-3}$ at altitude 3.1km. Like for the Stuttgart flight, the assimilation increases the concentrations of ash and the extent of the plume. The MOCAGE ash concentrations near Hamburg are around 20 $\mu$g.m$^{-3}$ in MOCAGE without assimilation, around 15 $\mu$g.m$^{-3}$ when mean lidar profiles are assimilated and reach 40 $\mu$g.m$^{-3}$ when maximum lidar profiles are assimilated (Fig. 11b).

At the same time the Ispra EARLINET lidar (Pappalardo et al., 2013) in the Po Valley detected ash at the altitudes 4 to 5 km. The MOCAGE with assimilation increases also the values of ash in this region. Although there is no quantitative estimate of ash concentrations, the range of values after assimilation increase after assimilation (Fig. 11d).

The assimilation of lidar data from two locations where an ash plume enters Europe induces corrections of ash concentrations as far as a thousand kilometres away over Europe. Although the flight measurements are sparse and have large error margins, the model estimates after assimilation compare more favourably to in-situ measurements than before assimilation. It is also noticeable that the correction by lidar cumulates in time: while the correction is rather low when the plume reaches Europe (17 May at 16UTC), it is larger in extent and intensity several hours after (18 at 10UTC). This may be due to the assimilation procedure, which has been done using a continuous hourly configuration.

## 5 Conclusions

This study investigated the benefit for the 3D representation of volcanic ash of a resolved source term and of the assimilation of different observations datasets, using the MOCAGE model. The main findings are:

- the use of a resolved source term instead of a parametrised source term induces a more realistic representation of the horizontal dispersion of the ash plume,

- a positive impact of the assimilation of MODIS AOD on the horizontal dispersion the plume has been shown, but this effect is rather low and local, compared to source term improvement,

- the continuous assimilation of lidar profiles from two ground-based stations improves the vertical distribution of ash and helps to simulate ash concentrations closer to those values obtained from in-situ observations.

As shown during the EUNADICS-AV project and demonstrations (Hirtl et al., 2019), a reliable representation of volcanic ash concentrations is needed to manage air traffic. The assimilation of lidar information is a way forward to tackle the tendency of model simulations to dilute ash in the vertical (Plu et al., 2021). Future work on other cases should confirm the results of the present study, before being able to apply them in an operational context.

A better resolved source term should have positive impacts on the vertical distribution of ash, and also on its grain size distribution. A perspective would be to assess how much these effects can change the optical properties of ash clouds and so the assimilation of data downstream. A better source term can also be beneficial as a better a priori for inversion of satellite column ash load.

The rather low impact of the assimilation of MODIS AOD on this case could be due to different reasons, one of them being the revisit time of polar-orbiting satellites and the possibility that it crosses an ash plume. The assimilation of AOD from geostationary satellites, such as MTG in the future (Descheemaecker et al., 2019), by increasing the time frequency of the measurements, could increase the impact of assimilation in space and time.

The present study is the first one, to our knowledge, that assesses the impact of continuous assimilation of ground-based lidars on a volcanic ash cloud. When the ash cloud reaches continental Europe, there is a clear benefit of assimilating lidar profiles to better constrain the concentrations of ash and their vertical distribution. Since 2010, there has been an increase of the density of lidars in Europe, and operational networks have been installed (operational lidars in France and in the United Kingdom, EUMETNET E-PROFILE network). Additionally the number of advanced lidars operating continuously within

ACTRIS/EARLINET has also increased. Based on our results, we can expect that, these data, if assimilated in aerosol transport models, can be highly beneficial for the 3D representation of ash concentrations.

This work opens new perspectives regarding the assimilation of lidar in dispersion models for volcanic ash monitoring and forecasting. Firstly, the processing of lidar data as input for assimilation requires some work: which lidar variable would be the most suited for assimilation? How to aggregate the values on a vertical scale to take into account the different resolutions of model and measurements? Secondly, the tuning of assimilation algorithms, depending on the input data, needs also to be done. In order to tune and achieve good quality assimilation of lidar for ash monitoring, there is a need for more observations on volcanic ash clouds, particularly for sampling the concentration of ash in-situ. The rarity of volcanic eruptions could be mitigated by studying volcanic clouds worldwide. It is worth also considering other high-concentration aerosol events, such as the dispersion of desert dust or of emissions from forest fires. Synthetic eruption may also be studied.

*Author contributions.* BS implemented FPLUME in MOCAGE. BS and EE designed the assimilation of aerosols. GB analysed the case study, designed the simulations and developed some diagnostics. LEA made some simulations. BJ and JG developed some aerosol components and helped with running MOCAGE. LM provided interpretation of EARLINET observations and lidar assimilation results. MP developed and generated most of the results, conducted their analysis and was the main writer.

*Competing interests.* The authors do not have competing interests.

*Financial support.*

This work has been conducted within the framework of the EUNADICS-AV project, which has received funding from the European Union's Horizon 2020 research programme for Societal challenges - smart, green and integrated transport under grant agreement no. 723986.

*Acknowledgements.* The authors thank A. Folch for having provided the FPLUME model for implementation in MOCAGE.

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

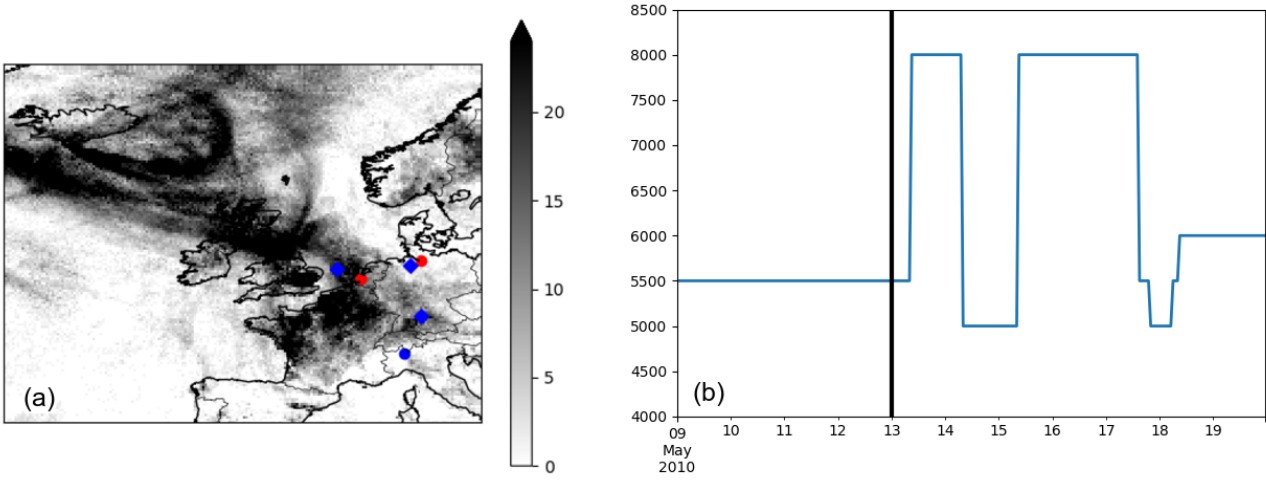

**Figure 1.** (a) Map of the number of times (at hourly step from 13 to 20 May 2010) when a column is contaminated by volcanic ash (ash column load above 0.2 g.m$^{-2}$) according to the observations (grey shadings). Diagnostics and scores are computed in this domain. The red (resp. blue) dots indicate the Cabauw and Hamburg (resp. Ispra) lidars. The blue diamonds refer to the location of the flight legs where aerosol measurements are available. (b) Emission height profile (m above sea level) used as input of the source term. The emission starts on 9 May in the model, but the evaluation of simulations starts on 13 May (vertical line).

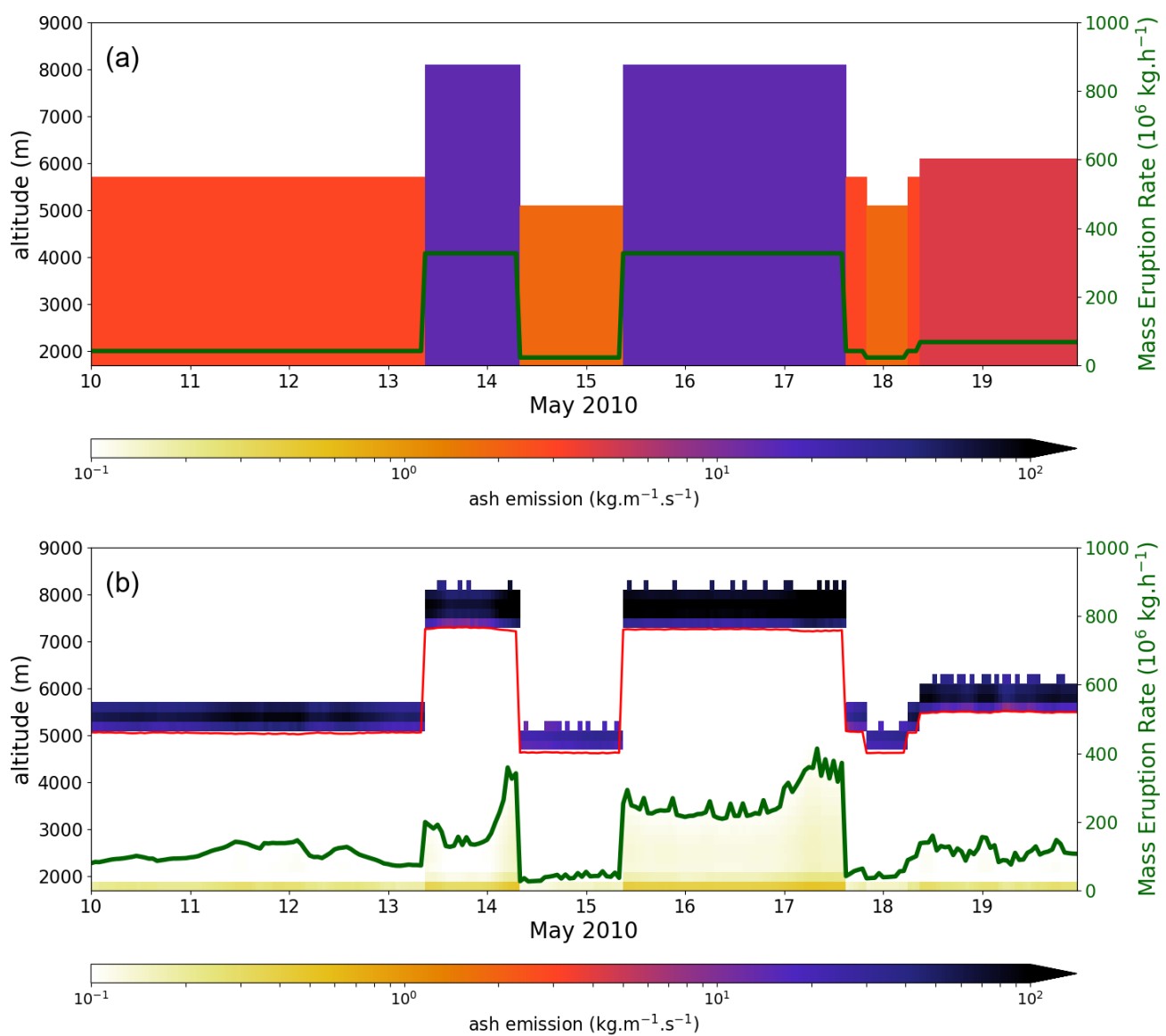

**Figure 2.** Comparison of MOCAGE ash source terms (unit: kg.m$^{-1}$.s$^{-1}$), as a function of time (horizontal axis) and altitude (left vertical axis), from 10 to 19 May 2010, for (a) the parametrised source term and (b) the source term resolved by FPLUME. The green lines and right vertical axis refer to the Mass Eruption Rate taken from the two source terms respectively. In the bottom plot (b), the red line shows the neutral buoyancy level that is computed by FPLUME.

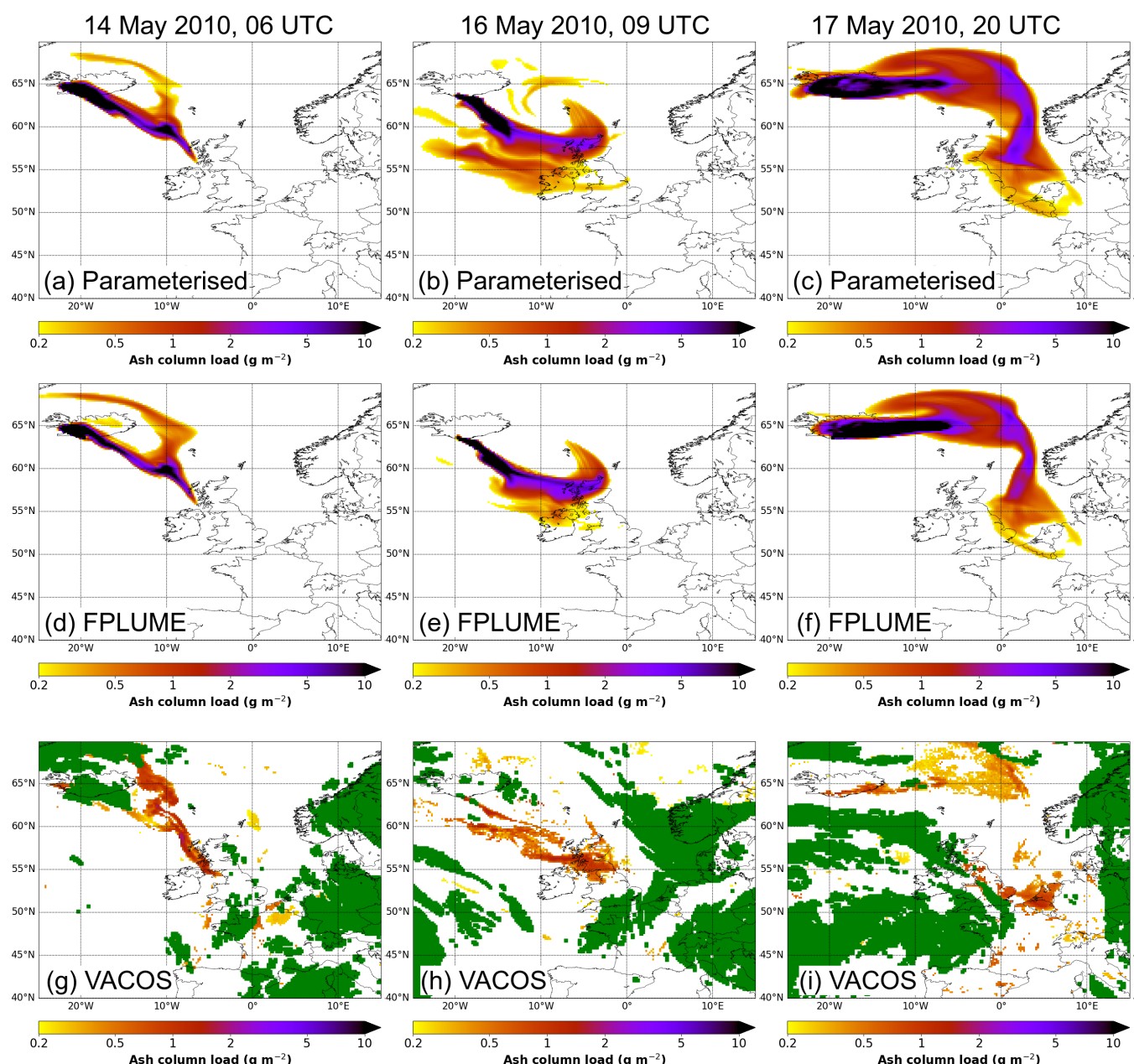

**Figure 3.** Total ash column simulated by MOCAGE using the parametrised source term (a, b, c) and the source term resolved by FPLUME (d, e, f), and estimated by the VACOS retrievals (g, h, i), on 14 May 2010 at 06 UTC (a, d, g), on 16 May 2010 at 09 UTC (b, e, h) and on 17 May 2010 at 20 UTC (c, f, i). The green colour refers to gridpoints where the ash retrieval could not be done (due to the presence of clouds, for instance).

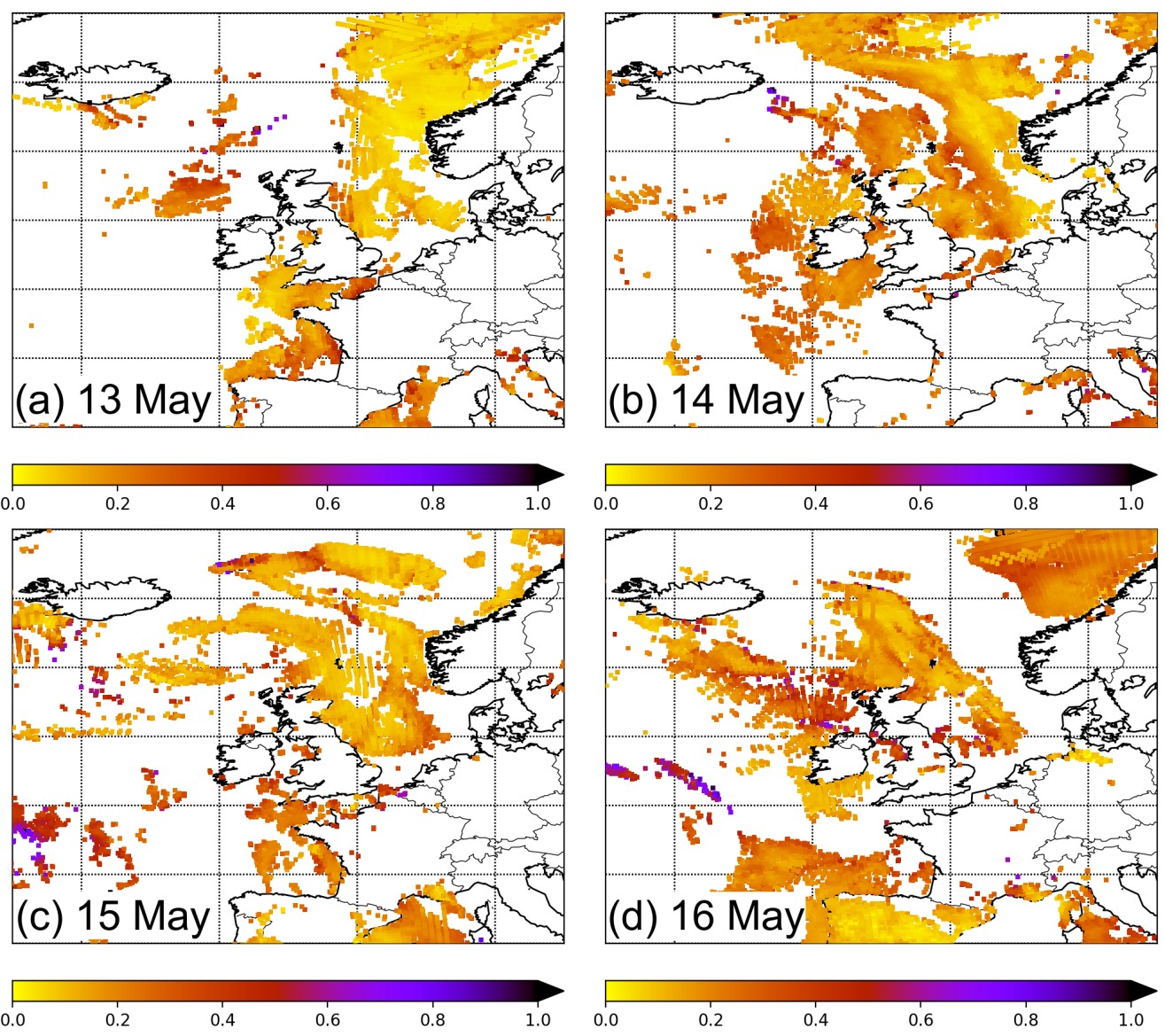

**Figure 4.** Daily values of AOD from TERRA and AQUA MODIS assimilated in MOCAGE, for (a) 13 May, (b) 14 May, (c) 15 May and (d) 16 May. The assimilated gridpoints are on a 0.2° resolution grid in order to match the MOCAGE grid. White areas are where no MODIS data is assimilated.

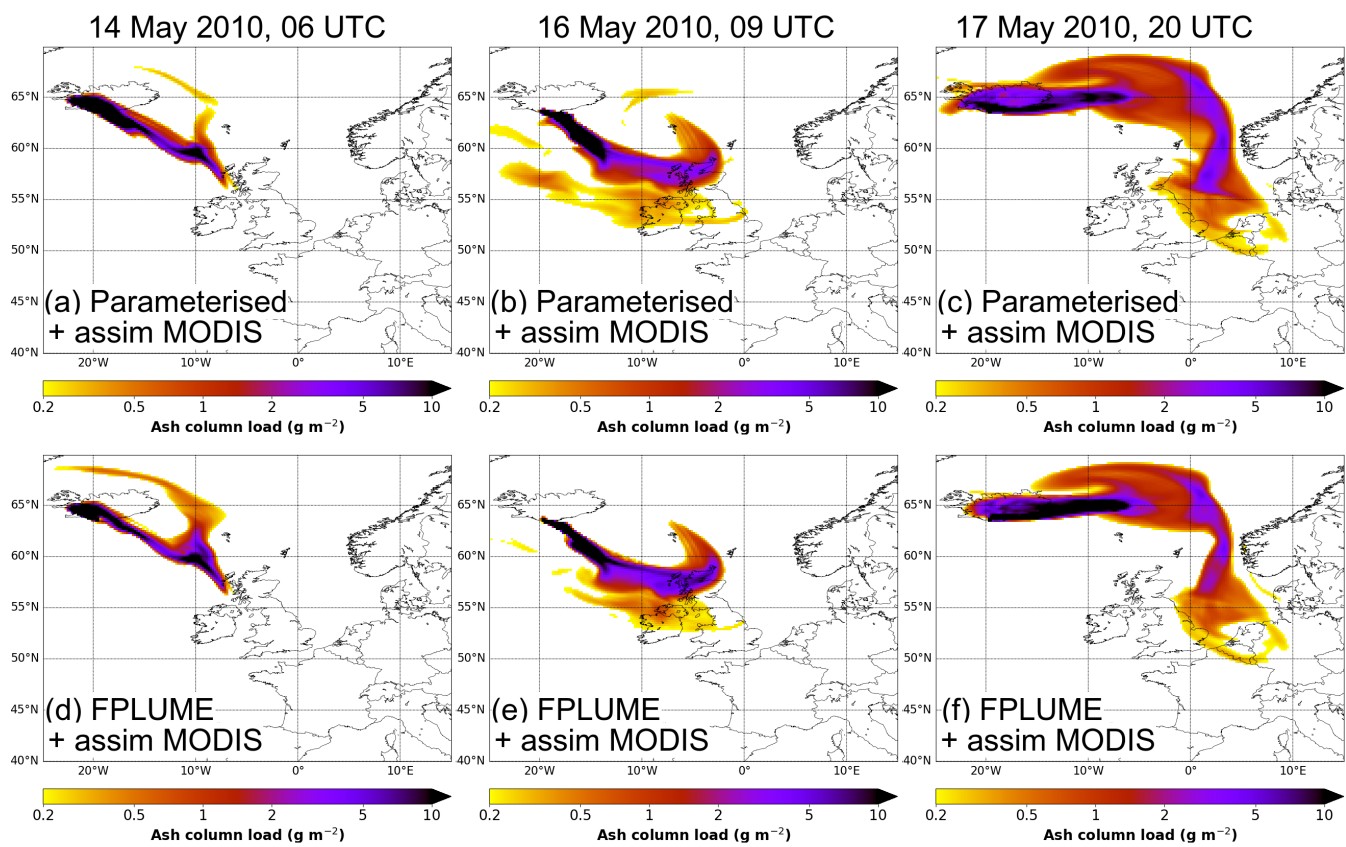

**Figure 5.** Same legend as Figure 3, for MOCAGE simulations after assimilation of MODIS AOD: using the parametrised source term (a, b, c) and the source term resolved by FPLUME (d, e, f), on 14 May 2010 at 06 UTC (a, d), on 16 May 2010 at 09 UTC (b, e) and on 17 May 2010 at 20 UTC (c, f).

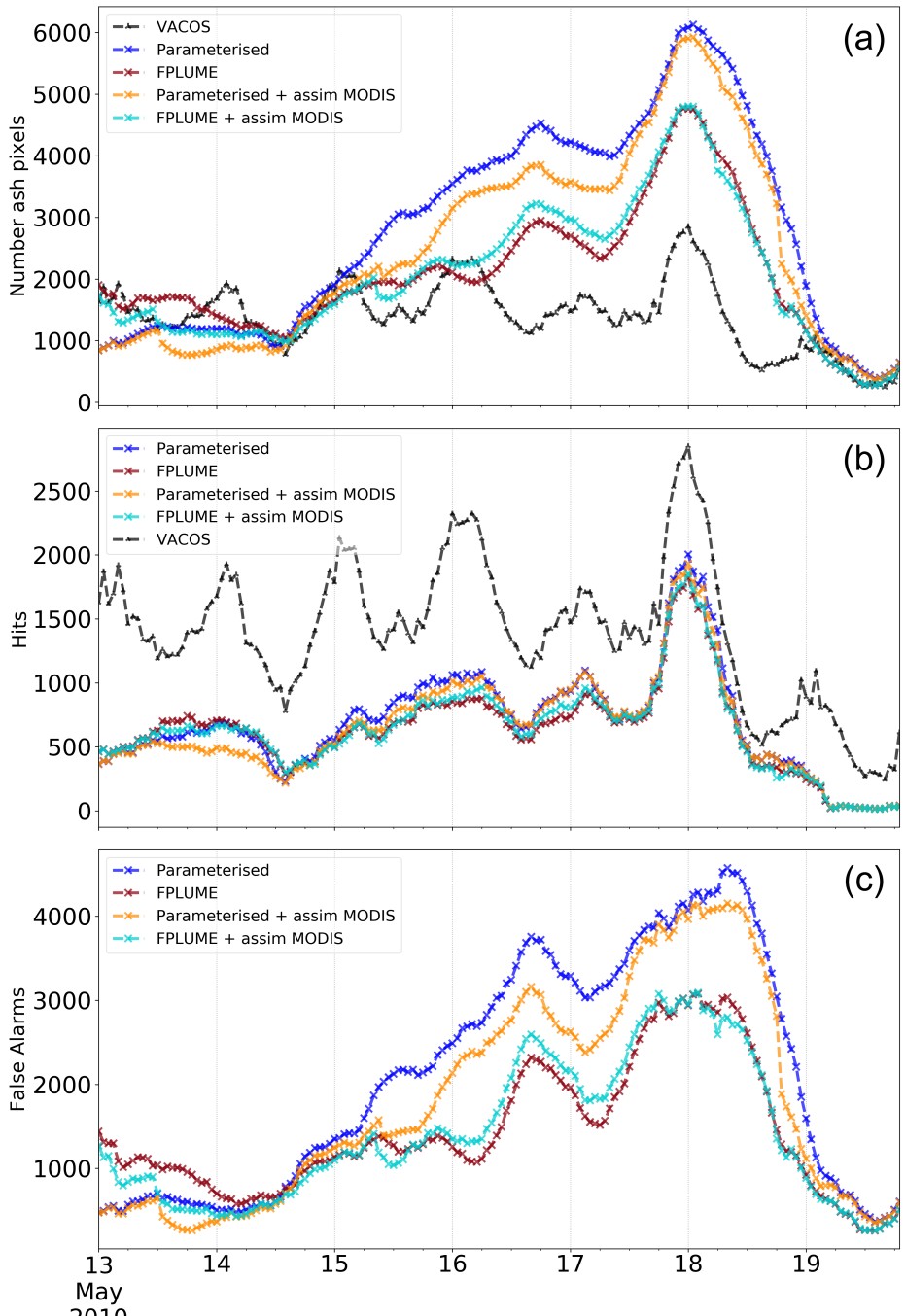

**Figure 6.** Comparison of the scores of ash contamination of the four MOCAGE simulations (with parametrised/FPLUME source term, with or without assimilation of MODIS AOD observations), against VACOS ash estimates. From top to bottom: (a) number of ash contaminated gridpoints in VACOS (black line) and in the MOCAGE simulations (colour lines)), (b) number of ash contaminated gridpoints in VACOS (black line) and also the number of hits (colour lines) for each simulation (gridpoints that are contaminated in both the simulation and in VACOS), and (c) number of false alarms for each simulation (gridpoints that are contaminated in the simulation and not in VACOS).

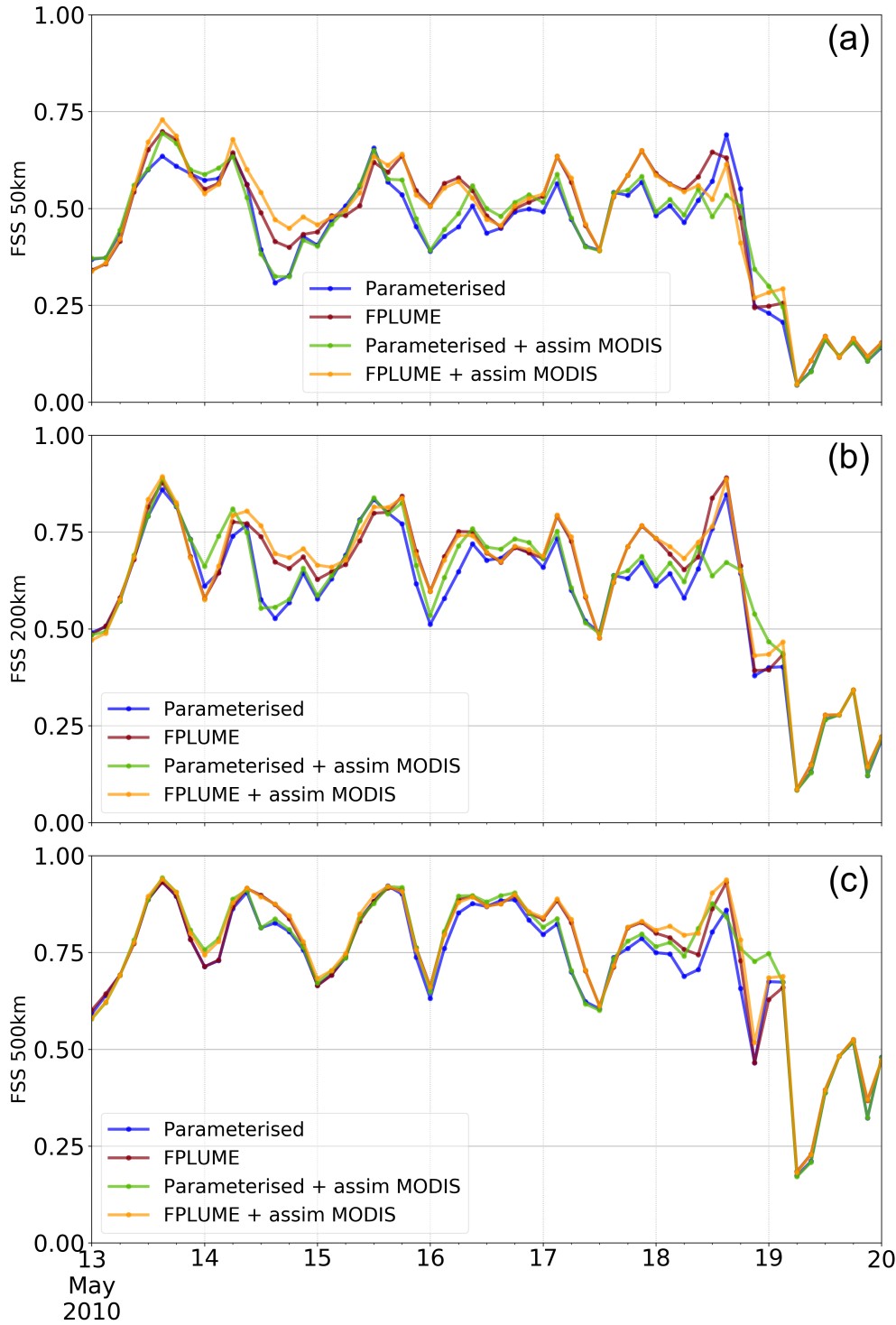

**Figure 7.** Comparison of the FSS of the four MOCAGE runs against VACOS estimates. The FSS values are shown for radii of (a) 50 km, (b) 200 km and (c) 500 km.

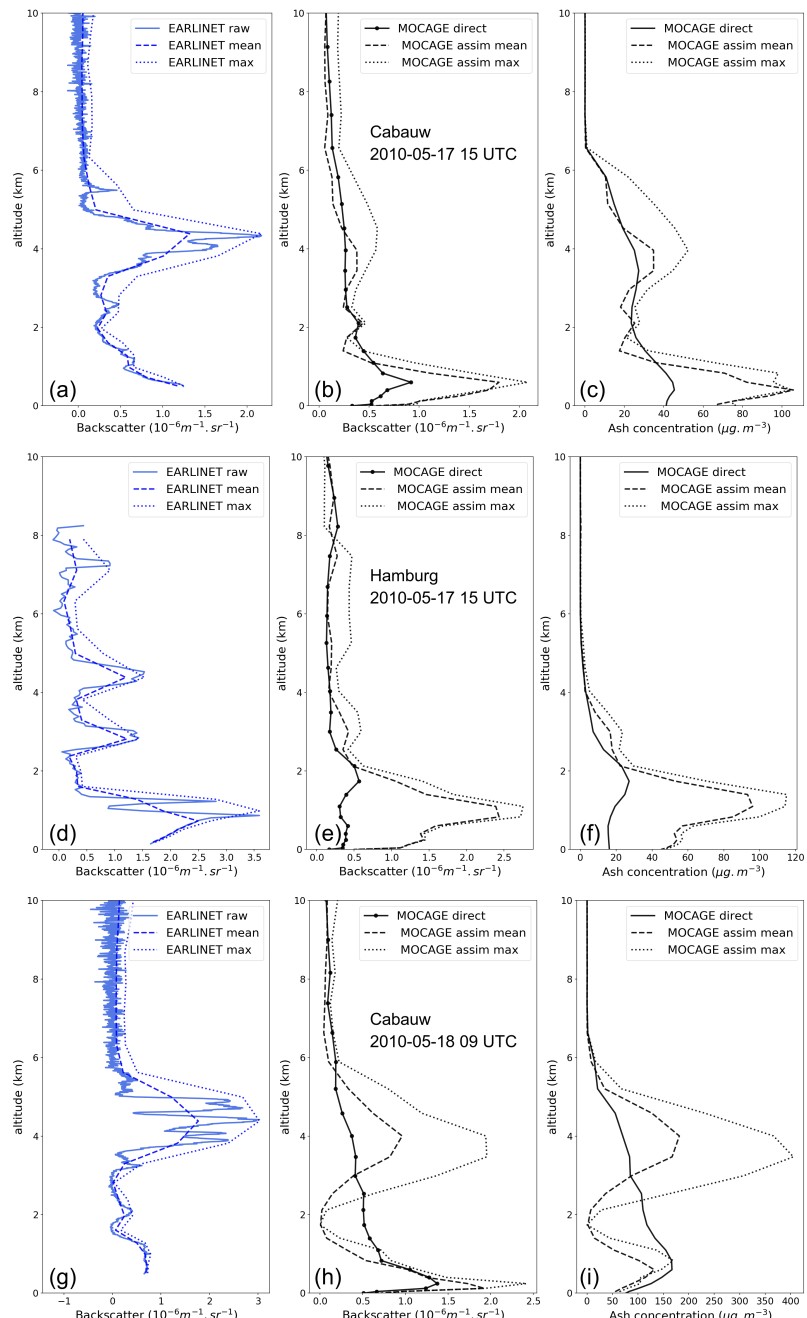

**Figure 8.** Vertical profiles of EARLINET backscatter coefficients (a, d, g), of MOCAGE backscatter coefficients (b, e, h) and of MOCAGE ash concentrations (c, f, i), at the Cabauw station on 17 May 2010 at 15 UTC (a, b, c), at the Hamburg station on 17 May 2010 at 15 UTC (d, e, f) and at the Cabauw station on 18 May 2010 at 9 UTC (g, h, i). From the original EARLINET profiles, mean or maximum values profiles are derived for assimilation in MOCAGE, at the vertical resolution of the model. The three MOCAGE simulations correspond to experiments without assimilation, with assimilation mean values, and with assimilation of the maximum values.

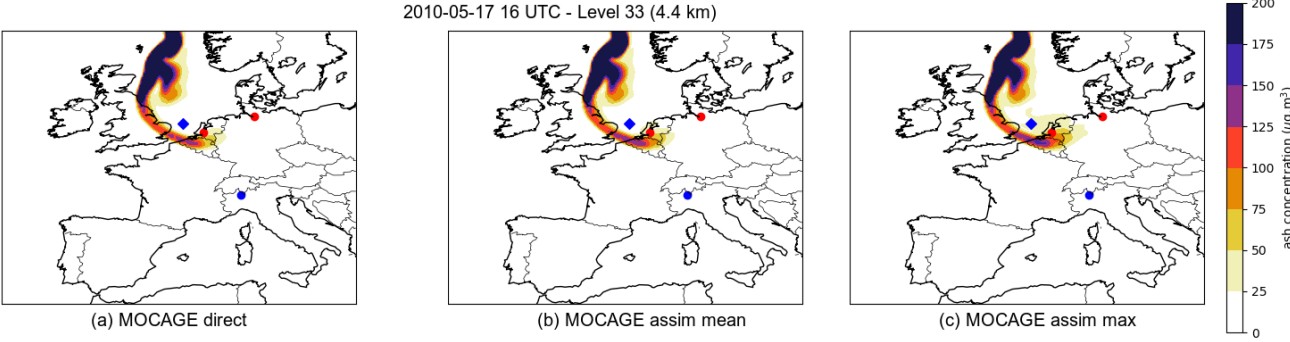

**Figure 9.** Comparison of volcanic ash concentration at level 33 (around 4.4 km), on 17 May 2010 at 16 UTC, simulated by (a) the MOCAGE simulation without assimilation, left panel), by (b) the MOCAGE simulation with assimilation of EARLINET mean profiles, and by (c) the MOCAGE simulation with assimilation of EARLINET max profiles. Ash concentration unit is $\mu$g.m$^{-3}$. The red dots indicate the Cabauw and Hamburg lidars, that are assimilated. The blue diamonds refer to the location of Flight 10 where aerosol measurements are available.

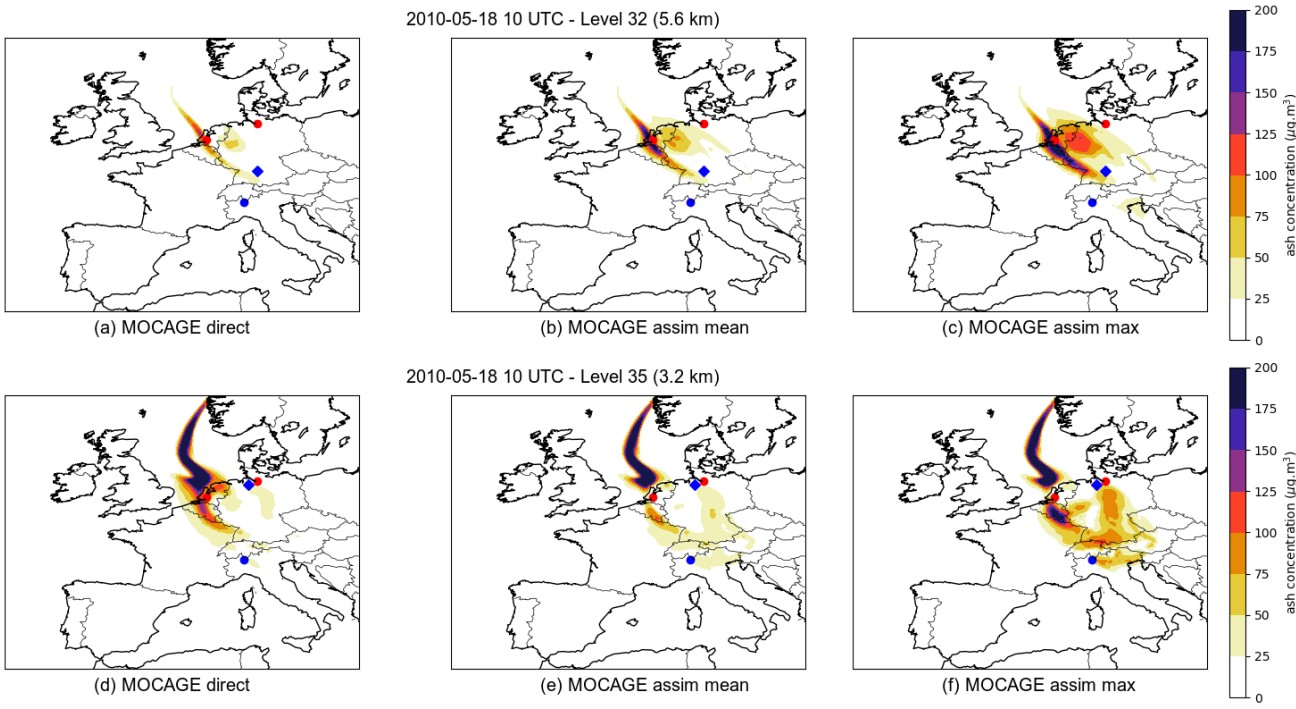

**Figure 10.** Same legend as Fig. 9, at level 32 (around 5.6 km, a, b, c), corresponding to Flight 12 (blue diamond, around Stuttgart) and at level 35 (around 3.2 km, d, e, f), , corresponding to Flight 11 (blue diamond, around Hamburg) on 18 May 2010 at 10 UTC.

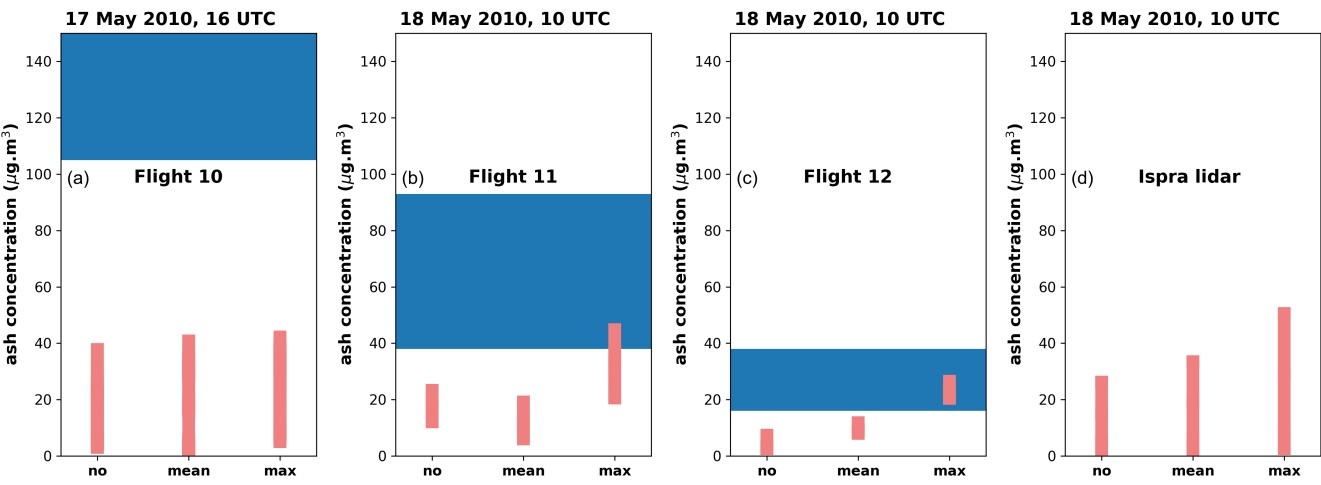

**Figure 11.** Comparison of ash concentration (vertical axis, unit $g\,m^{-3}$), for the three MOCAGE simulations (no assimilation, mean-values lidar assimilation and maximum-values lidar assimilation, from left to right in each panel), at the location of the measurements: (a) flight 10 on 17 May at 16 UTC, (b) flight 11 on 18 May at 10 UTC, (c) flight 12 on 18 May at 10 UTC, and (d) Ispra lidar on 18 May at 10 UTC). The ranges of in-situ flight measurements are shown as blue horizontal rectangles. For the MOCAGE data (red bars), the values of several gridpoints are plotted, that sample the ash concentration at locations that correspond to the measurements