# Peer review of "Modelling the volcanic ash plume from Eyjafjallajökull eruption (May 2010) over Europe: evaluation of the benefit of source term improvements and of the assimilation of aerosol measurements"

_Natural Hazards and Earth System Sciences, 2021_

## Author Comment (AC1)

**RC1**

The authors thank the reviewer for his positive evaluation of the manuscript and for his comments about the text. Here are the answers to the Reviewer Comments 1 (RC1) and how they have been addressed to improve the manuscript.

*The methods used are in general sound, and the paper is, except for some problems with the use of the English language, well written and organized.*
Use of the English language has been improved after a careful proofreading and rewriting of sentences that are too long or unclear.

Major issues:

*I think the vertically constant distribution of mass in the Mastin formulation for the source term is quite unrealistic. It is well known that volcanic ash clouds are usually umbrella-shaped. Thus, the unrealistic assumption of a vertically constant ash distribution puts the Mastin formulation at a somewhat unfair disadvantage in the comparison with FPLUME. Assuming a generic shape with a maximum close to the observed plume heights would be much more realistic, and may even outperform FPLUME. I see, of course, the value of using FPLUME, but often these models create a lot of variability that may not match well reality, and a smoother but still somewhat realistic profile may actually lead to better results.*

A complementary simulation has been done, in which the source term follows a vertical umbrella shape. In this simulation, for a total mass M that is emitted and a plume height H, the mass that is emitted per meter follows a profile that is linear between the vent height (at which emission is zero) and 0.73H, and parabolic between 0.73H and H. 0.75M is emitted in the upper parabolic section, and 0.25M is emitted in the linear section. M and H are related with the Mastin et al (2009) et al relationship, with the same assumptions that are used in the manuscript for the "Parameterised" simulation.

The maps and scores of this new simulation ("Parameterised umbrella") are shown in the following figures. They are compared to the simulations that are presented in the manuscript: the "parameterised" one, renamed here as "Parameterised uniform" and the "FPLUME" one.

This new "umbrella" simulation generates more ash load than the two other simulations, which already overestimated ash load. Accordingly, the number of contaminated gridpoints (above 0.2 g.m2 ash load) increases, and the "hit score" detection for ash contaminated areas is higher, at the expense of having more false alarms. The FSS of FPLUME is higher than the results of the other simulations, most of the time. From these results, we cannot state that introducing an umbrella-shaped source term instead improves the simulation with a uniform vertical repartition of ash nor outperforms the FPLUME simulation.
We propose to add in the manuscript that we have tested an umbrella-shape source term, but not to add significant graphics about this simulation, which does not bring new information to the manuscript, in our opinion.

[Figure]

Figure - Total ash column simulated by MOCAGE using the parameterised source term with a vertical uniform distribution of mass (upper panels), the parameterised source term with an umbrella shape (middle panels) and the source term resolved by FPLUME (bottom panels), on 14 May 2010 at 06 UTC (left panels), on 16 May 2010 at 09 UTC (middle panels) and on 17 May 2010 at 20 UTC (right panels).

[Figure]

Figure - Same legend as Figure 4 of the manuscript, for the three MOCAGE simulations: using the parameterised source term with a vertical uniform repartition of mass (blue lines), the parameterised source term with an umbrella shape (red lines) and the source term resolved by FPLUME (yellow lines).

[Figure]

Figure - Same legend as Figure 6 of the manuscript, for the three MOCAGE simulations: using the parameterised source term with a vertical uniform repartition of mass (blue lines), the parameterised source term with an umbrella shape (red lines) and the source term resolved by FPLUME (green lines).

*Line 172: How does the variation of fine ash from 0.1 to 5% come about in FPLUME? Shouldn't the size distribution be more a function of the eruption properties, rather than processes in the atmosphere? The latter (e.g., aggregation) may also have some effect but I am surprised these results in such large variations, and I am wondering how realistic these are.*

The fraction of fine ash varies from 0.1% à 5% one vertical level to another and one time to another, so it is not surprising to find such large variations. If we look at the fraction of fine ash from one instant to another, the variation is from 0.4% to 0.9%, which is a lower range. We propose to change the sentence to: "is dispersed (i.e. which size falls into the fine ash classes and will be introduced in MOCAGE) varies from 0.4\% to 0.9\%, depending on time."

At the entry of FPLUME, a constant distribution of mass is supposed at the vent, which is the same as the one for the parameterized source term. FPLUME solves atmospheric processes that modifies the ash mass per bins and along the vertical dimension. At the output of FPLUME, ash is distributed into bins between particles of diameter from 61 nm up to 64 mm. These descriptions have been added in the new manuscript.

*You are assimilating MODIS data but you never show them. However, this is essential for the reader to understand the effect of the MODIS data assimilation. It is surprising that the MODIS data assimilation has so little effect, so this needs some more discussion as well.*

[Figure]

Figure – Representation of the assimilated AOD values from MODIS in MOCAGE, from 13[th] to 16[th] May. Each map represent daily values, but assimilation in MOCAGE is done at a hourly step. In the white areas, no AOD value was taken into account in the assimilation during this day.

From the figure above that presents the AOD values that are assimilated in MOCAGE, it is obvious that many assimilated AOD gridpoints are close to the ash plume and even some of the high AOD values belong to the plume (particularly on 14 and 16 May). We propose to add this figure in the manuscript in order to support the discussion about the impact of assimilation of MODIS AOD on this event.

*The comparison to the aircraft data is somewhat disappointing. A scatter plot of observed vs simulated values would be much more convincing than just the mentioning of a few values in the text.*

An additional figure has been prepared, that shows, for the different measurements discussed in the manuscript, the observed values and the values of the simulations based on the different MOCAGE assimilation simulations (no assimilation, assimilation of mean lidar values, assimilation of maximum lidar values).

[Figure]

Figure - Comparison of ash concentration (vertical axis, unit g\,m$^{-3}$), for the three MOCAGE simulations ("no" assimilation, "mean"-values lidar assimilation and "max"imum-values lidar assimilation, in abscissa), at the location of the measurements (from left to right: flights 10 on 17 May at 16 UTC, flight 11 on 18 May at 10 UTC, flight 12 on 18 May at 10 UTC, and Ispra lidar on 18 May at 10 UTC). The range of in-situ flight measurements are plotted as blue horizontal rectangles. For the MOCAGE data (red bars), the values of several gridpoints are plotted, that sample the ash concentration at locations that correspond to the measurements.

This figure helps to assess the quantitative benefit of the assimilation of the lidar profiles, even quite far away to the location of the lidar instruments. It helps to support the interpretation of the results. We propose to add this figure in the manuscript, and to revise the discussion in a more quantitative manner.
Besides, the concentration maps (Figures 8 and 9 of the manuscript) have been completed by the location of the Ispra EARLINET lidar.

*The figures are generally well presented; however, the labels in figures 3 and 7 are MUCH too small. On a print-out they are totally unreadable.*
These figures has been done with larger label fonts.

Minor issues:

*I find it surprising that all simulations are so similar in terms of their FSS values shown in Fig. 6. Does that mean that FSS is not a particularly sharp measure to evaluate the performance of the model?*
The FSS values of our experiments are generally in a similar range as other models as shown in another published study by Plu et al (2021, NHESS). Still, from the arguments provided in the manuscript and in the literature (Harvey and Dacre, 2016), a higher FSS means that a better location of the most intense patterns of ash plumes.

*Line 267, last word: I think this should be right, not left, column*

Yes, done.

*Line 270: What are vertical processes?*
" insufficient vertical resolution, grid-scale vertical velocity, diffusion, aerosol sedimentation" have been added to the manuscript.

*Figure 1: Clarify what exactly is shown with number of grid points contaminated with ash. Does this mean the number of vertical levels?*
To clarify, the legend has been modified : " Map of the number of times (at hourly step from 13 to 20 May 2010) when a column is contaminated by volcanic ash (ash column load above 0.2~g.m$^{-2}$)…"

Language:
*line 9: dispersion OF the plume*
*line 12: hundred kilometers downstream of*
*quite often singular and plural are mixed, e.g. line 26: forecasts REMAIN a challenge; line 52: source termS; line 57: performance IS compared; line 124: of the componentS of ACTRIS; line 127: each of the aerosol layerS; line 192: ash pOcket DOES not show up*
*line 57: are used IN the case study*
These language errors have been corrected.

We hope that we have addressed RC1 satisfactorily and that, after implementation of theses changes in the manuscript, it can be accepted for publication.

---

## Author Response (AR1)

Dear Editor,
 please find here, as requested, the new version of the manuscript after the reviews.
Best wishes,
M. Plu